# Direct observation of topological surface-state arcs in photonic metamaterials

Biao Yang[1], Qinghua Guo[1,2], Ben Tremain[3], Lauren E. Barr[3], Wenlong Gao[1], Hongchao Liu[1], Benjamin Béri[1], Yuanjiang Xiang[2], Dianyuan Fan[2], Alastair P. Hibbins[3] & Shuang Zhang[1]

The discovery of topological phases has introduced new perspectives and platforms for various interesting physics originally investigated in quantum contexts and then, on an equal footing, in classic wave systems. As a characteristic feature, nontrivial Fermi arcs, connecting between topologically distinct Fermi surfaces, play vital roles in the classification of Dirac and Weyl semimetals, and have been observed in quantum materials very recently. However, in classical systems, no direct experimental observation of Fermi arcs in momentum space has been reported so far. Here, using near-field scanning measurements, we show the observation of photonic topological surface-state arcs connecting topologically distinct bulk states in a chiral hyperbolic metamaterial. To verify the topological nature of this system, we further observe backscattering-immune propagation of a nontrivial surface wave across a three-dimension physical step. Our results demonstrate a metamaterial approach towards topological photonics and offer a deeper understanding of topological phases in three-dimensional classical systems.

[1] School of Physics and Astronomy, University of Birmingham, Birmingham B15 2TT, UK. [2] Key Laboratory of Optoelectronic Devices and Systems of Ministry of Education and Guangdong Province, College of Optoelectronic Engineering, Shenzhen University, Shenzhen 518060, China. [3] Electromagnetic and Acoustic Materials Group, Department of Physics and Astronomy, University of Exeter, Stocker Road, Exeter EX4 4QL, UK. Biao Yang, Qinghua Guo and Ben Tremain contributed equally to this work. Correspondence and requests for materials should be addressed to A.P.H. (email: A.P.Hibbins@exeter.ac.uk) or to S.Z. (email: s.zhang@bham.ac.uk)

Topology, describing invariant properties under continuous deformation, is known to govern the physics of macroscopic systems. Topologically nontrivial systems, including topological insulators[1, 2], topological superconductors[2], Dirac semimetals[3, 4], Weyl semimetals[5–9], and topological photonics/phononics[10–18], have been proposed and studied both in theory and experiment owing to their exotic behaviors. Reflecting the emergence of materials with novel massless fermions beyond fundamental particle analogues, Weyl semimetals are classified as type-I or type-II[19]. For a type-I Weyl semimetal, the Weyl point at the Fermi level exhibits a point-like Fermi surface. Whereas for a type-II Weyl semimetal, the Weyl point emerges from a point-like contact between an electron and a hole pocket, signifying a highly tilted Weyl cone and a Lifshitz transition in the associated Fermi surface. On the interface of a bulk sample, gapless surface states exist, protected by chiral topological charges associated with the Weyl points. These topological surface states, which take the form of Fermi arcs connecting the projections of bulk Fermi surfaces containing Weyl points on the surfaces, show characteristic signatures of the topological nature of Weyl semimetals.

Several works have reported experimental observations of Fermi arcs for both type-I and type-II Weyl points[8, 9, 20, 21] in solid state systems by using angle resolved photo emission spectroscopy (ARPES) or scanning tunnelling microscopy (STM) measurements. In photonics, point degeneracy of a photonic type-I Weyl node has been reported in a microwave angle-resolved transmission measurement of a double gyroid photonic crystal[22]. Thereafter, a number of groups proposed the realization of Weyl degeneracies in other classical systems[23–26]. However, as a very important signature of Weyl degeneracies, the topological surface-state arc (TSA) has not yet been directly observed in classical systems.

Here, we report the experimental observation of photonic TSA at the interface between a photonic type-II Weyl metamaterial

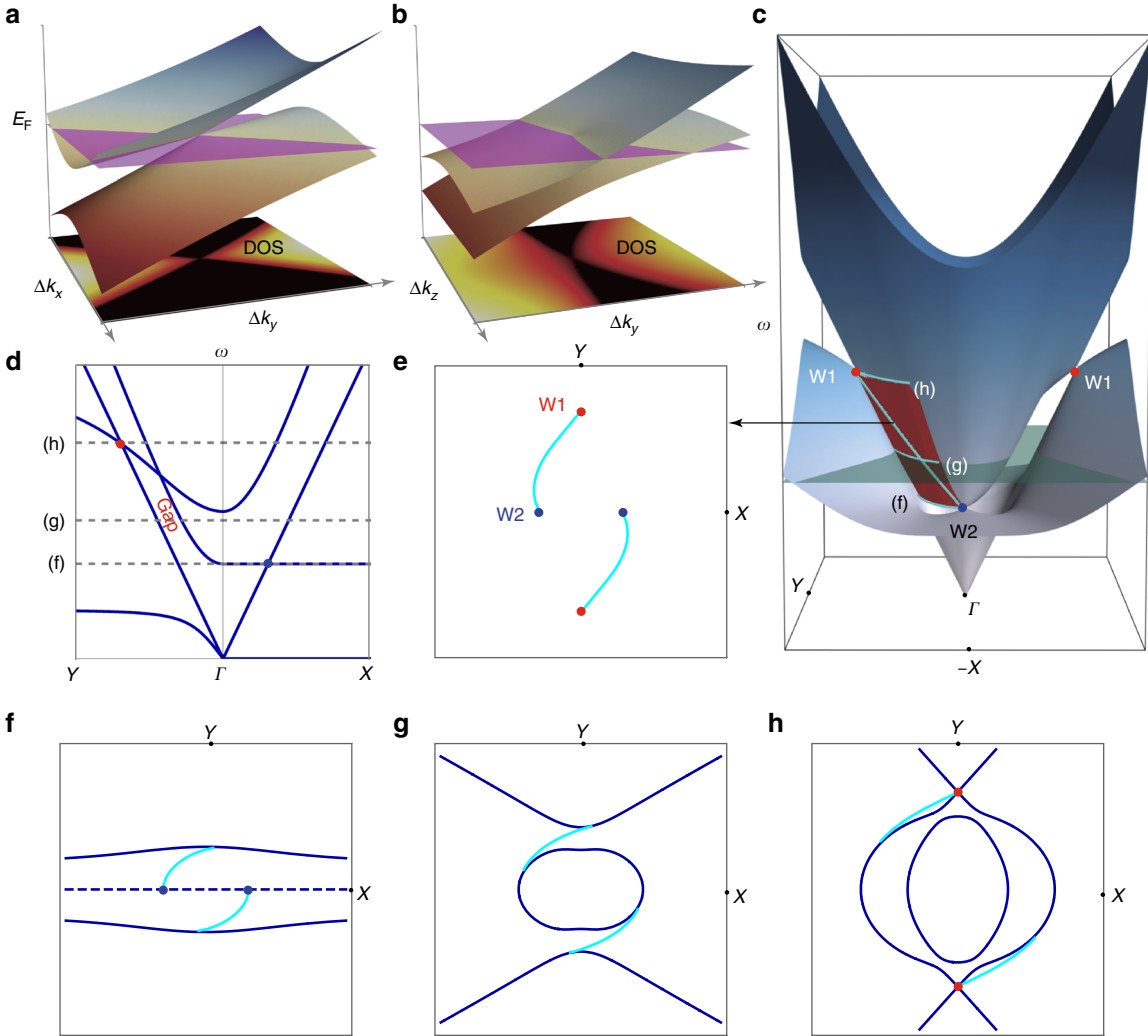

**Fig. 1** Bulk and surface states in an effective-medium model of chiral hyperbolic metamaterials. Schematic view of energy dispersion near a type-II Weyl point for the chiral hyperbolic metamaterials (CHM) with respect to **a** $x$–$y$ and **b** $y$–$z$ momentum space. *Bottom planes* show momentum resolved density of states (DOS) at the 'Fermi energy' $E_F$ indicated by the *purple plane*. **c** Effectively modeled band structures of bulk (7th and 8th bands) and surface states for the CHM. *Red dots* (W1) indicate one pair of type-II Weyl points. *Blue dots* (W2) (one is hidden) are their chiral partners. One surface state between two topological partners is indicated by the *red surface*, on which the *cyan lines* highlight surface-state arcs. To show the gap, the other surface state is not plotted; it can be obtained after time-reversal operation of the present one. **d** Bulk bands along $Y(0,−3,0)$–$\Gamma(0,0,0)$–$X(3,0,0)$. **e** Surface band structure on a varying-frequency $k_x$–$k_y$ map. The arc ($k_y > 0$) is constructed from intersection between the surface state (*red surface* in **c**) and a plane with constraints: $\omega$ is proportional to $k_y$ and $k_x$ is arbitrary. **f–h**, show three equi-frequency contours (EFCs) containing both bulk and surface states on the $k_x$–$k_y$ plane. Their corresponding frequencies are indicated in **c** with *cyan lines* and in **d** with *dashed lines*, respectively. In **f**, the *dashed line* indicates the flat band ($\Gamma$–$X$) in **d**. Here, $\varepsilon_0 = \mu_0 = c = 1$, $\omega$, $k_x$, and $k_y$ are normalized with respect of $\omega_p$ (at $k_y = 0$), which is the plasma frequency

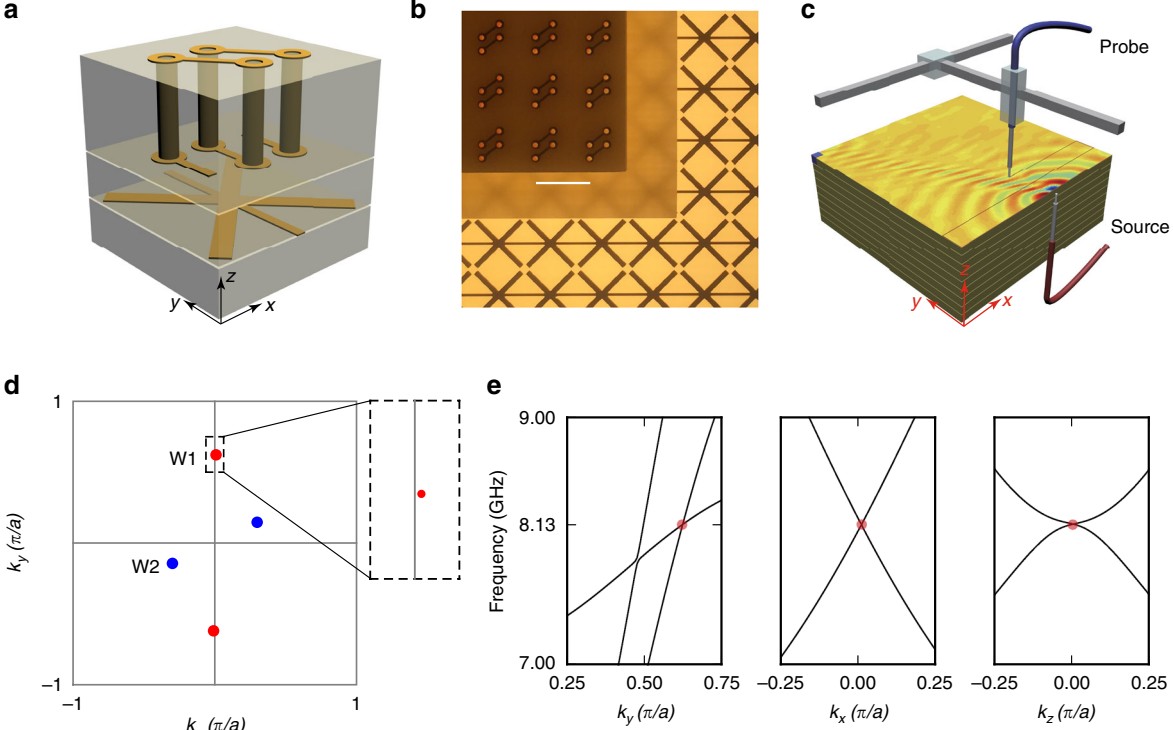

**Fig. 2** Photonic type-II Weyl nodes realized within a chiral hyperbolic metamaterial. **a** Schematic of the cubic unit cell of chiral hyperbolic metamaterial (CHM) with side length $a = 4$ mm, consisting of three layers; bottom hyperbolic layer (thickness = 1 mm), middle blank layer (thickness = 1 mm), and top chiral layer (thickness = 2 mm). Each helix has 2.5 turns, with its axis along the x-direction. Its length and cross-sectional area are optimized to give a fundamental resonance frequency around 5 GHz. The blank layer (FR4) between the chiral and hyperbolic layer is designed to avoid shorting contact between them. **b** Tri-layer sample fabricated with printed circuit board technology. There are 75 unit cells ($4 \times 4$ mm$^2$) along the in-plane directions on each layer. On the hyperbolic layer, metallic lines go through the whole layer along the y-direction. The *scale bar* indicates 4 mm. **c** Experimental setup and layer-stacking geometry. The field pattern represents the real experimental data scanned under x (polarized source) − z (polarized probe) configuration at 5.82 GHz. The surface wave is excited by one antenna, whilst another serves as a probe scanning the propagating near field. **d** The slice of the first Brillouin zone with respect of $k_z = 0$ and locations of Weyl points for W1: ($\pm 0.01, \pm 0.62$) $\pi/a$ and W2: ($\pm 0.30, \pm 0.14$) $\pi/a$. The zoomed-in rectangle indicates that W1 is located slightly away from the $k_y$ axis. **e** Simulated linear degeneracy around type-II Weyl point W1 (8.13 GHz) with the realistic structure designed in CST Microwave Studio

and air. Through Fourier transformation of the measured real-space electromagnetic field at the interface, we directly obtain the TSA in the momentum space, which is shown to connect between two topologically distinct bulk states. Furthermore, our measurements directly map out the surface wave propagation across a step-shape cut out from the metamaterial, serving as a direct visualization of the robustness of topological phase.

## Results

**Design of the topological photonic metamaterial.** To observe the characteristics of a Weyl semimetal, a sample that possesses either broken time-reversal symmetry or broken inversion symmetry is required. In classical electromagnetism, it is more feasible to break inversion symmetry, as breaking time-reversal symmetry requires lossy magnetic materials and external magnetic fields. Here a chiral hyperbolic metamaterial (CHM) with broken inversion symmetry, which was recently proposed as a topological metamaterial[27], is employed to explore the photonic TSA. The topological nature of the metamaterial can be described by a homogeneous effective model and its Weyl points arise from the degeneracies between intrinsic electromagnetic modes: the longitudinal plasmonic mode and the spin-polarized transverse mode[27, 28]. It is distinct from typical photonic realization of Weyl degeneracies in photonic crystals[22, 24, 26], where spatial degrees of freedom span the state sub-space. The two bands that cross each

other forming the Weyl cone have the same sign of velocity along a certain momentum space direction ($k_y$ as shown in Fig. 1a, b). In a simple effective model (Supplementary Note 1) of the CHM, one pair of type-II Weyl points[27, 28], represented by the two *red dots* (W1) in Fig. 1c, is located at large $k$ and therefore becomes easier to identify experimentally. Without any mirror symmetries, the other pair of Weyl points, represented by two *blue dots* W2 (only one in shown) at a lower frequency in Fig. 1c are chiral partners of the two W1 Weyl points. Both W1 (W2) Weyl nodes are related to each other by time-reversal symmetry and have the same chirality. Nontrivial surface states between the Weyl point partners (W1 and W2) are also shown in Fig. 1c (*red surface*), which serve as a signature of the topological nature of the system. The band structure along high symmetry lines and the nontrivial gap (as indicated in *red*) where the topological surface states reside are shown in Fig. 1d. In Fig. 1e, TSA connecting W1 and W2 Weyl points with varying frequency are projected onto the $k_x$–$k_y$ plane. Its three-dimensional view can be found in Fig. 1c as the tilted *cyan line*. To demonstrate the evolution of the surface states with frequency, three equi-frequency contours (EFCs) are shown in Fig. 1f–h. In Fig. 1f, one clearly sees TSA originating from the W2 Weyl points, whose bands along ΓX are flat (*dashed line*), behaving like a transition phase between type-I and type-II Weyl points[25]. As the frequency increases, TSA will be tangent to the corresponding bulk states, which contain the projections of Weyl points. Finally, they terminate at W1 in Fig. 1h.

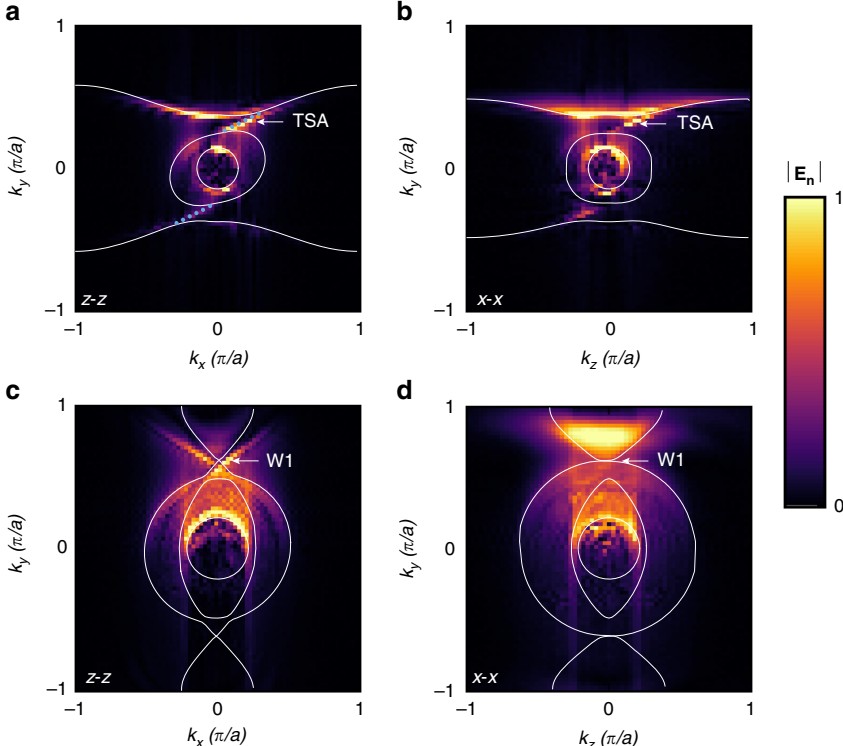

**Fig. 3** Photonic equi-frequency contours at two different frequencies. Scanned electric field with the configuration of $z$ (polarized source) − $z$ (polarized probe) on the top surface at **a** 5.46 GHz and **c** 8.13 GHz. Scanned results with the configuration of $x$ (polarized source) − $x$ (polarized probe) on the side surface at **b** 5.46 GHz and **d** 8.13 GHz. For $x$ polarized source, the antenna is oriented in the −$x$ direction. In each the innermost *solid circle* indicates the light cone. The *other solid curves* present simulated bulk states of the chiral hyperbolic metamaterial from CST Microwave Studio. Data in the $k_y < 0$ range are back-scattered from the edge of the sample. Topological surface-state arcs (*TSA*) and type-II Weyl points (W1) are indicated by the white arrows. In **a** the *dotted cyan lines* indicate simulated TSA. The crossing points in **c**, **d** show another characteristic feature of type-II Weyl points as shown in Fig. 1a and b density of states (DOS) planes. In all *panels*, the normal components of the electric fields are normalized to the same reference value

The CHM is constructed by stacking of a two-dimensional tri-layer unit. Figure 2a illustrates a cubic unit cell of the CHM with a period $a = 4$ mm, which is fabricated on copper-clad FR4 substrates with dielectric constant of 4.1. The thickness of each copper layer is 35 µm and the copper can be regarded as a perfect electric conductor (PEC) in the studied frequency range 5–10 GHz. To obtain the desired hyperbolic properties of the CHM, 200 µm-wide metallic wires are formed along the $y$-direction on the top surface of the bottom layer (Fig. 2b). Metallic crosses are superimposed on these wires to increase the local capacitance and suppress the strong non-local effects induced by the metallic wires alone[29]. To break inversion symmetry, we introduce a metallic helix structure on the top layer. Owing to the small number of turns per unit length, electric current driven along the helix induces a magnetic dipole moment that is slightly misaligned with the $x$-axis[30]. This leads to a small shift ($0.01\pi/a$) of W1 away from the $k_y$ axis, and a larger shift ($0.14\pi/a$) of W2 away from $k_x$ axis as shown in Fig. 2d. Nevertheless, they are still located on the $k_z = 0$ plane owing to both the time-reversal symmetry and $C_2$ rotational symmetry along the $z$-axis. The large momentum separation of the Weyl points, as shown in Fig. 2d, supporting the existence of a very long TSA, allows us to easily resolve TSA from the Fourier transformation of near-field scanning. Here, we would like to emphasize that although the existence of TSA between a pair of opposite Weyl points is guaranteed by the bulk-surface correspondence, its exact location depends on the electromagnetic properties of the surrounding medium through the boundary conditions of Maxwell's equations. For the realistic structure, Fig. 2e shows simulated band crossing in the vicinity of the type-II Weyl node (W1). The simulated dispersion around W2 is presented in Supplementary Fig. 2.

**Direct observation of photonic topological surface-state arcs.** The near-field scans are first conducted on both top ($xy$) and side ($yz$) surfaces of the CHM with co-polarized source and probe, that is, $z$ (polarized source) − $z$ (polarized probe) for the top surface scan, and $x$ (polarized source) − $x$ (polarized probe) for the side. After Fourier transformation of the near-field spatial distribution, we obtain Fig. 3a and b presenting the EFC (both bulk and surface states) at 5.46 GHz for the top and side surfaces, respectively. As can be seen, TSA are tangent and terminated to their corresponding bulk modes in both top and side cases. Figure 3c and d shows the top-scanned and side-scanned EFC at 8.13 GHz, respectively. From Fig. 3c, one can see a resemblance to the modelled density of states (DOS) in Fig. 1a, a feature that provides verification of type-II Weyl dispersion. As previously discussed, the misalignment of the helix axis with the underlying lattice results in a shift of the W1 Weyl points away from the $k_y$ axis. However, this shift is too small to be observed in our experiment due to limited $k$-space resolution with a minimum pixel size of $0.027\pi/a$. In Fig. 3d, a similar DOS crossing can be seen from the side surface scan, as schematically shown in Fig. 1b. It is worth noting that W2 Weyl points (5.15 GHz) appear accompanied by a strong resonance from the metallic helices and cannot be clearly recognized in the experiment. These measured bulk/surface band structures are consistent with the simulations

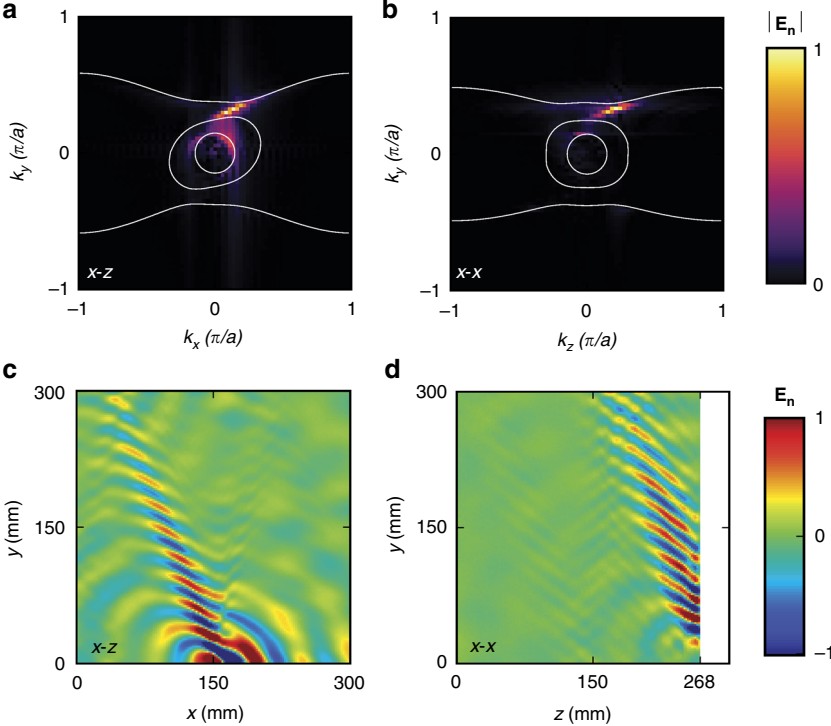

**Fig. 4** Momentum and real space representations of topological surface-state arcs. **a**, **c** Top surface scan taken with the $x$ (polarized source) − $z$ (polarized probe) configuration at 5.46 GHz. **c** Real-space instantaneous field. **a** Momentum space amplitude field after Fourier transformation of **c**. **b**, **d** Side surface scan with respect of $x$ (polarized source) − $x$ (polarized probe) configuration at 5.46 GHz. **b**, **d** Same plot as **a** and **c**, but on the side surface. The source position for the side surface scan is on the top surface with coordinate ($x = 10$ mm, $y = 0$ mm, and $z = 268$ mm). *Solid circles* indicate the light cone, and are far away from topological surface-state arcs, thus the surface wave is well localized. The normal components of the electric fields in **a**–**d** are normalized to the same reference values, respectively

(from CST Microwave Studio) in terms of the TSA and their locations in momentum space, as shown in Fig. 3. The deviations between the measurements and calculations regarding the exact shape and size of each TSA may arise from the sample fabrication errors and misalignment of layer stacking. Additional comparisons between experiment and simulation results at different frequencies are given in Supplementary Fig. 3.

In general, the scanned near-field intensity profile can vary significantly with the measurement configuration, such as the proximity to the sample surface and the polarizations of source and probe, which offers great potential for topological near-field sensing. Our near-field scanning measurements on the top and side surface configurations with different polarization combinations can provide complementary information for revealing the topological features of CHM. For instance, Figure 4a and c shows a surface wave on the top surface with $x$ (polarized source) − $z$ (polarized probe) configuration. Both real space and momentum spaces show the strong excitation of surface states in the nontrivial gap by the antenna source. The intensity contribution from bulk bands, on the other hand, is largely suppressed and decays rapidly away from the source position ($x = 150$ mm, $y = 0$ mm). When the surface wave impinges on a surface edge, which is invariant along the $y$ direction, the topologically nontrivial surface wave is expected to travel around the right-angle edge and continues to propagate on the side surface ($y$–$z$ plane). Figure 4d shows the results when an $x$-polarized probe scans the side surface with the source placed on the top surface ($x = 10$ mm, $y = 0$ mm, $z = 268$ mm). One can observe that the bulk modes are markedly weakened by edge backscattering but the surface wave is not, which can also be confirmed by its bright TSA as shown in Fig. 4b compared with Fig. 3b.

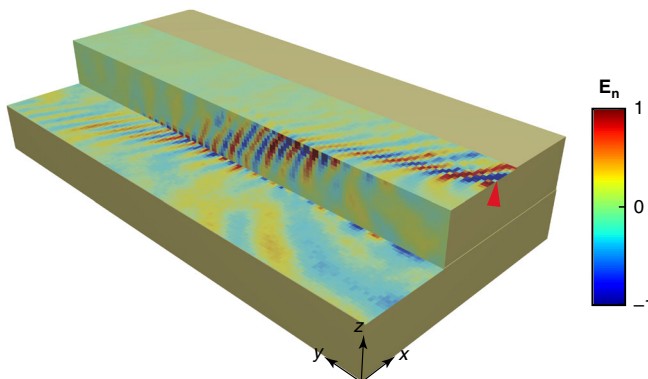

**Fig. 5** Topologically protected surface waves propagating on a step. A backscattering-immune surface wave propagates on a three-dimensional step geometry. The source (*red triangle*) is set with $x$-polarization, whereas the normal component of the electric field is probed for all surfaces. The step width, height, and length are 104 mm (both for upper and lower surfaces), 60 and 600 mm, respectively. The normal electric field is normalized with respect to its maximum value. (See real experimental setup in Supplementary Fig. 1).

**Topologically protected surface wave steps down.** To demonstrate the topological protection of the surface states, a step configuration is created by stacking a metamaterial block on top of another with a 104 mm shift in the $x$ direction, as shown in Fig. 5. A surface wave is launched at the upper surface, and the probe scans the normal field distribution in the upper, side, and lower surfaces at a frequency of 5.46 GHz. It is observed that the

surface wave conformally bends around the step and continues propagating forward without being reflected by the edges. It should be noted that since our probe antenna cannot reach the lower corner (built by side and lower surfaces) owing to a thick cladding layer of the antenna, there exists a discontinuity in the measured phase across the lower corner. This measurement of robust propagation of a surface wave across the step serves as a direct observation of the topological protection of TSA.

## Discussion

The metamaterial approach, described by an effective model, is relatively simple in the designs and realizations of different topological phases. Without the constraints of high dielectric-constant and strict periodicity, as required in photonic crystals, it is also more feasible to be fabricated and assembled. Furthermore, owing to the advantage of deep sub-wavelength in metamaterials, surface waves are usually well-confined on the interface. Thus, near-field scanning greatly facilitates the measurements and discoveries of novel surface states in various topological photonic metamaterials. Although metallic elements are very lossy in the optical band, the same design can be transferred to terahertz systems, where a nonmagnetic terahertz TSA hopefully can be observed.

In conclusion, we have made a direct observation of photonic TSA at the surfaces of a photonic type-II Weyl metamaterial. Our work may well provide new insights and open new avenues to surface photonics. Owing to the robustness of topologically protected surface-wave propagation and other practical advantages in photonic systems, such as artificially controllable structure design, we anticipate our observation to be a starting point for surface periscope imaging technology, near-field sensing, and directional information transmission in bulky integrated photonics.

## Methods

**Numerical calculations**. Numerical calculations are performed using the eigenmode solver in CST Microwave Studio. Both FR4 substrate and PEC are considered as lossless materials in the simulation. Bulk states are calculated in a single unit cell configuration accompanied with periodic boundary conditions. Supercell, consisting of 20 unit cells, is used to simulate surface states. All CST files are available from the corresponding authors on request.

**Near-field scanning setup**. The experimental setup is schematically shown in Fig. 2c, instead of using an angle-resolved transmission measurement, we employ a microwave vector network analyser (VNA) and a near-field antenna acting as a source (stationary) to provide excitation of electromagnetic surface waves, which are subsequently probed with a second near-field antenna (controlled by an $xyz$ translation stage). The band structure of the surface states can then be determined from Fourier analysis of the spatial distribution of the electric field at each frequency. In the scanning measurements, the scan step, which is set to 2 mm, equaling half the lattice constant of the cubic unit cell ($4 \times 4 \times 4$ mm$^3$), will determine the maximum surface $k$-space range as $[-2\pi/a, 2\pi/a]^2$ under Fourier transformation. With fixed scan step, the $k$-space resolution is controlled by the maximum area being scanned, that is, the size of sample fabricated. Here, each unit cell can be considered as a sub-wavelength electromagnetic meta-atom with a period of approximately $\lambda/10$. The collective effect of a large number of these meta-atoms is measured, which is the key principle of metamaterials in general. Because the type-II Weyl point is located far away from the light cone, the topological surface wave is well-confined at the interface between the CHM and air, which greatly facilitates the near-field measurements.

**Source and probe antennas**. The measurement is performed in the microwave regime using a VNA to sweep the frequency (5–10 GHz), with a near-field antenna connected to the excitation/detection port. The antenna consists of a coaxial cable with a length (1 mm) of outer conductor and sheath stripped away, leaving the central conductor (with core diameter 1 mm) exposed, which can provide efficient coupling to large-momentum surface modes owing to the broad range of momenta present in the exponentially decaying near field both for bulk and surface states. The antenna also provides a method of measuring the amplitude and phase of the near field of the modes with sub-wavelength resolution. The probe is most sensitive to field components parallel to its orientation.

**Sample fabrication and assembly**. The sample is fabricated by a commercial printed circuit board (PCB) company. Two-sample assemblies for different measurements are shown in Supplementary Fig. 1. To clearly extract bulk and surface states from the CHM, a 300 mm$^3$-block is built, which is assembled by 75 periods of tri-layers along the $z$ direction. Each tri-layer consists of three sub-layers: chiral (2 mm-thick), blank (1 mm-thick), and hyperbolic layer (1 mm-thick). We scanned both top surface and side surface with probe and source antennas as shown in Supplementary Fig. 1a. Both antennas can be bent to orient them in arbitrary directions to meet our different polarization requirements. Supplementary Fig. 1b shows the step configuration for verifying back scattering-immune propagation. In total, four blocks are used to construct the surface wave. The interface in the middle may introduce slight scattering for the surface wave. In the measurement, we scanned normal components of three different surfaces (top, side, and lower) with probe step of 4 mm.

**Data availability**. The data that support the findings of this study are available from the corresponding author on request.

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

## Acknowledgements

We thank Ling Lu for discussions and feedback. This work was financially supported by ERC Consolidator Grant (Topological), the Royal Society and the Wolfson Foundation. B. Y. acknowledges China Scholarship Council (201306110041). S.Z. and B.B. acknowledge support from the Royal Society. Y.X. and D.F. acknowledge support from the National Natural Science Foundation of China (Grant Nos. 61490713 and 61505111). Q.G. acknowledges support from the National Natural Science Foundation of China (Grant No. 11604216). L.E.B., B.T. and A.P.H. acknowledge financial support from the Engineering and Physical Sciences Research Council (EPSRC) of the United Kingdom, via the EPSRC Centre for Doctoral Training in Metamaterials (Grant No. EP/L015331/1). All data created during this research are openly available from the University of Exeter's institutional repository at https://ore.exeter.ac.uk/repository/handle/10871/28162.

## Author contributions

The data were measured and analyzed by B.Y., Q.G., B.T., L.E.B., A.P.H., and S.Z. The sample was designed by B.Y. and S.Z. The experimental concept was developed by A.P.H., B.T., and S.Z. All authors contributed extensively to the discussion of the results, as well as to the preparation of the manuscript.

## Additional information

**Competing interests:** The authors declare no competing financial interests.

