## [Peer Review File · Nature Communications]

REVIEWERS' COMMENTS:

Reviewer #1 (Remarks to the Author):

Authors presented their experimental results on the observation of Fermi arcs and the corresponding topological protection of surface states. The experiment follows the previous work by Soljacic's group, where they observed a very weak signature of Weyl points. The work is a nice demonstration of Weyl point physic. I recommend for publication in Nature Comm.

It would be nice if the authors could comment on where the allowed momenta of the arc are expected. This discussion was not clear from their PRL theory publication.

REVIEWERS' COMMENTS:

Reviewer #1 (Remarks to the Author):

Authors presented their experimental results on the observation of Fermi arcs and the corresponding topological protection of surface states. The experiment follows the previous work by Soljagic's group, where they observed a very weak signature of Weyl points. The work is a nice demonstration of Weyl point physic. I recommend for publication in Nature Comm.

It would be nice if the authors could comment on where the allowed momenta of the arc are expected. This discussion was not clear from their PRL theory publication.

Our reply:

We thank the reviewer for his/her positive view of our work. The existence of Fermi arc surface states is guaranteed by the bulk-surface correspondence condition. Although the Fermi arcs must connect the projected opposite Weyl points, or the Equipfrequency Frequency Contours that enclose the Weyl points, the allowed momenta (the exact location of Fermi arc in the momentum space) are determined by the boundary condition of electromagnetic wave, and therefore they are different for different surrounding media. This is illustrated by the plot of the Fermi arcs at the interface of a hyperbolic chiral medium with air and perfect electric conductor (PEC) as the surrounding media, respectively. Note that for simplicity, the calculation is performed on an effective medium with constitutive relations defined as,

$$\boldsymbol{\varepsilon} = \begin{bmatrix} 3 & 0 & 0 \\ 0 & 3 & 0 \\ 0 & 0 & -3 \end{bmatrix}, \boldsymbol{\mu} = \begin{bmatrix} 1 & 0 & 0 \\ 0 & 1 & 0 \\ 0 & 0 & 1 \end{bmatrix}, \boldsymbol{\gamma} = \begin{bmatrix} 1 & 0 & 0 \\ 0 & 0 & 0 \\ 0 & 0 & 0 \end{bmatrix}$$